# Central Regulation of Brown Fat Thermogenesis in Response to Saturated or Unsaturated Long-Chain Fatty Acids

**DOI:** 10.3390/ijms24021697

**Published:** 2023-01-15

**Authors:** Anna Fosch, Maria Rodriguez-Garcia, Cristina Miralpeix, Sebastián Zagmutt, Maite Larrañaga, Ana Cristina Reguera, Jesus Garcia-Chica, Laura Herrero, Dolors Serra, Nuria Casals, Rosalia Rodriguez-Rodriguez

**Affiliations:** 1Basic Sciences Department, Faculty of Medicine and Health Sciences, Universitat Internacional de Catalunya, 08195 Sant Cugat del Vallès, Spain; 2INSERM, Neurocentre Magendie, U1215, University of Bordeaux, 3300 Bordeaux, France; 3Department of Biochemistry and Physiology, School of Pharmacy and Food Sciences, Institut de Biomedicina de la Universitat de Barcelona (IBUB), Universitat de Barcelona, 08028 Barcelona, Spain; 4Centro de Investigación Biomédica en Red de Fisiopatología de la Obesidad y la Nutrición (CIBEROBN), Instituto de Salud Carlos III, 28029 Madrid, Spain

**Keywords:** long-chain fatty acids, hypothalamus, thermogenesis, brown adipose tissue, CPT1C, obesity

## Abstract

Sensing of long-chain fatty acids (LCFA) in the hypothalamus modulates energy balance, and its disruption leads to obesity. To date, the effects of saturated or unsaturated LCFA on hypothalamic-brown adipose tissue (BAT) axis and the underlying mechanisms have remained largely unclear. Our aim was to characterize the main molecular pathways involved in the hypothalamic regulation of BAT thermogenesis in response to LCFA with different lengths and degrees of saturation. One-week administration of high-fat diet enriched in monounsaturated FA led to higher BAT thermogenesis compared to a saturated FA-enriched diet. Intracerebroventricular infusion of oleic and linoleic acids upregulated thermogenesis markers and temperature in brown fat of mice, and triggered neuronal activation of paraventricular (PaV), ventromedial (VMH) and arcuate (ARC) hypothalamic nuclei, which was not found with saturated FAs. The neuron-specific protein carnitine palmitoyltransferase 1-C (CPT1C) was a crucial effector of oleic acid since the FA action was blunted in CPT1C-KO mice. Moreover, changes in the AMPK/ACC/malonyl-CoA pathway and fatty acid synthase expression were evoked by oleic acid. Altogether, central infusion of unsaturated but not saturated LCFA increases BAT thermogenesis through CPT1C-mediated sensing of FA metabolism shift, which in turn drive melanocortin system activation. These findings add new insight into neuronal circuitries activated by LCFA to drive thermogenesis.

## 1. Introduction

The regulation of metabolism and food intake in response to nutrients is tightly controlled by the hypothalamus, which detects nutrient availability via direct metabolic signals and, in turn, adjusts appetite, energy expenditure, and peripheral metabolism [1,2,3]. Disturbances in the sensing of or response to metabolic and nutritional signals in the hypothalamus are on the basis of metabolic diseases such as obesity and type 2 diabetes [4,5]. Despite that lipids play a minimal role as energy substrates in neurons [6], long-chain fatty acids (LCFA) and intermediates of fatty acid (FA) metabolism act as signaling molecules of nutrient surplus in the hypothalamus to regulate energy balance and the etiology of obesity [7]. There is strong evidence that sensing and fluctuations of FA synthesis precursors in the hypothalamus, such as malonyl-CoA, regulates whole-body energy metabolism [8,9]. In addition, central infusion of the unsaturated FA oleic acid leads to acute reduction of food intake, body weight and glucose production [1,10,11]. These oleic acid effects are closely related to intracellular accumulation of LCFA-CoA [11,12] and activation of the melanocortin system with increased expression of the anorexigenic and body weight loss promoter neuropeptide pro-opiomelanocortin (POMC) in the hypothalamus [10]. In contrast, central administration of saturated FA, highly consumed in Western diets, promotes body weight gain and increases local inflammation, leading to a dysfunction of critical neurons involved in the response to satiety and adipostatic signals [13,14]. In particular, saturated LCFA act predominantly through hypothalamic TLR4 and PKCƟ pathways to increase hypothalamic expression of pro-inflammatory cytokines (i.e., TNFα and IL1β) in microglia and endoplasmic reticulum (ER) stress as a downstream event promoting insulin and leptin resistance [13,14,15]. Central injection of unsaturated LCFA, in contrast, elicits a potent anti-inflammatory effect with increased expression of IL10 and IL6 in the hypothalamus [13,16]. The obesogenic potential of diets enriched in saturated LCFA was also related to lower activation of diet-induced thermogenesis in patients with obesity compared to unsaturated fats, without modifications in satiety responses, indicating that the degree of saturation of LCFA affects fat accumulation by altering energy expenditure [17]. Despite these findings, the exact molecular mechanisms involved in the hypothalamic sensing of different FA (saturated vs. unsaturated) to modulate peripheral metabolism has been poorly explored.

In response to high-fat feeding, obesity progression is associated with specific changes in the dynamic of lipids in the hypothalamus, particularly during the first days of diet administration [18,19,20]. These changes correlate with the activation of brown adipose tissue (BAT) thermogenesis as a key process to counteract excessive body weight gain [19,21,22]. BAT is considered a major site for energy burning in response to diet via sympathetic innervation, and it is primarily controlled by the hypothalamus and the brainstem [21,22,23,24]. Targeting the hypothalamic regulation of BAT thermogenesis is one of the most recent and promising strategies to manage energy balance [25]. In the last decade, intensive research has been directed toward the identification of neuronal targets and pathways in the hypothalamus involved in energy sensing to control BAT thermogenesis and in turn to counteract obesity development and progression [26]. However, the specific mechanisms mediating BAT thermogenesis in response to LCFA sensing and metabolism in the hypothalamus are still unknown.

A recently revealed neuronal protein involved in hypothalamic regulation of diet-induced BAT thermogenesis is carnitine palmitoyltransferase 1C (CPT1C) [27]. The CPT1 proteins are responsible for LCFA metabolism and are considered the main downstream effectors of the energy balance regulatory role of malonyl-CoA [8,28,29]. The canonical CPT1 isoforms (CPT1A and CPT1B) are ubiquitously expressed in different tissues and mediate FA oxidation (FAO) in the mitochondria. CPT1C, in contrast, is almost exclusively expressed in neurons and is located in the ER with negligible catalytic activity but is still able to bind the lipid intermediary malonyl-CoA, indicating its role as a nutrient sensor in neurons [8,28]. CPT1C is particularly found in the appetite regulatory nuclei arcuate (ARC), paraventricular (PaV) and ventromedial hypothalamus (VMH), and is crucial in leptin and ghrelin feeding response [30,31]. In fasting conditions, CPT1C senses the negative energy balance and controls fuel partitioning in the liver and muscles [32,33], whereas in response to fat-rich diets, CPT1C-KO mice present an obesogenic phenotype and insulin resistance [33,34]. We recently demonstrated that CPT1C deficiency leads to impaired activation of BAT thermogenesis following metabolic challenges such as short-term high-fat diet (HFD) exposure or central leptin infusion [27]. We also found that the canonical pathway AMPK(VMH)-Malonyl-CoA-sympathetic nervous system (SNS)-BAT-mediated thermogenesis was blunted in CPT1C-KO mice [27], suggesting CPT1C as a key player of the hypothalamic AMPK/ACC pathway in the control of BAT thermogenesis.

Considering the importance of LCFA-dependent signaling in hypothalamic regulation of energy balance and the etiology of obesity, the aim of this study is to explore the central effects of LCFA on BAT thermogenesis, taking into account the degree of saturation of the FA chain. We also investigate the specific hypothalamic nuclei involved and the implication of the neuronal nutrient sensor CPT1C in FA-induced BAT thermogenesis activation.

## 2. Results

### 2.1. Short-Term Administration of Monounsaturated (MUFA) or Saturated FA (SFA)-Enriched HFD Differently Affects Body Weight, Feeding, BAT Thermogenesis and Neuronal Activation in the Hypothalamus

To analyze the differential effects of saturated and unsaturated LCFA on the control of energy balance, wild-type (WT) mice were exposed to fat-rich diets with different fat compositions. In particular, animals were fed a diet with high content in monounsaturated FA (MUFA diet), saturated FA (SFA diet) or a standard diet (SD) used as a control compared to the HFD. In the first experimental approach, diets were administered for a short-term period of 7 days. This timing was based on our previous study showing the importance of these first days to adapt energy balance in response to a high-fat feeding [27].

After 7 days of administration of these diets, mice fed a SFA diet, but not a MUFA diet, showed higher body weight gain compared to the control SD group (Figure 1A). Despite the lack of significant effects of the MUFA diet on body weight change, this type of HFD led to an increase in both cumulative food (Appendix A) and calorie intake (Figure 1B) at different stages of the treatment, compared to the other experimental groups.

The analysis of BAT thermogenesis in response to the diets revealed that interscapular BAT (iBAT) temperature and mRNA expression of thermogenic markers were upregulated in MUFA diet-fed mice compared to SD (Figure 1C–E). Although the SFA diet group also showed a slight upregulation of genes related to thermogenesis in BAT vs. control mice, this induction was significantly lower in comparison to the MUFA group (Figure 1C). Taken together, these results suggest that MUFA diet-fed mice delayed body weight gain probably due to a higher activation of BAT thermogenesis even though animals are eating more calories, in contrast to that observed in SFA diet-fed mice.

Considering the crucial role of the hypothalamus in the regulation of energy balance, neuronal activation in specific nuclei of the hypothalamus was explored in mice fed with the different diets for 2 h. Evaluation of neuronal activation through c-Fos expression showed that neurons in the PaV were activated by the MUFA diet but not by the SFA diet, in comparison to the control group, whereas no changes were appreciated in the other nuclei analyzed (Figure 1F,G). This result reveals that the essential role of a MUFA diet in integrating hypothalamic signals mainly relays on PaV.

### 2.2. Intracerebroventricular (ICV) Injection of Unsaturated FA, but Not Saturated FA, Reduces Food Intake, Body Weight and Activates BAT Thermogenesis

In order to explore the specific action of LCFA present in the diets, unsaturated and saturated FA were injected ICV in mice. ICV administration of the unsaturated FA, oleic and linoleic acid, for 48 h reduced body weight and food intake change compared to vehicle-treated mice (Figure 2A–C), whereas the saturated FA, palmitic and stearic acid, had no significant effects on these parameters. The satiating action of oleic acid was supported by a notable increase in the hypothalamic expression of the anorexigenic neuropeptide POMC without changes in the expression of the orexigenic neuropeptide Y (NPY) and Agouti-related peptide (AgRP) neuropeptides in ICV oleic acid-treated mice (Appendix A).

The impact of central administration of FA on BAT thermogenesis showed a significant upregulation of thermogenic genes in brown fat of oleic and linoleic acid-treated mice (Figure 2D,E). In contrast, most of these mRNA levels remained unchanged or even reduced after central injection of the saturated FA under study (Figure 2F,G). The activation of BAT thermogenesis in response to unsaturated FAs was also supported by iBAT temperature measurement, showing an increase only with oleic and linoleic acids (Figure 2H,I).

### 2.3. ICV Injection of Oleic Acid, but Not Palmitic Acid, Increases Neuronal Activation in Hypothalamic PaV, VMH and ARC Nuclei

c-Fos immunofluorescence was assessed in different hypothalamic nuclei of mice 2 h after ICV administration of oleic and palmitic acid, mimicking MUFA and SFA diets experiments, respectively. After the ICV injection, mice did not have access to food. Results showed an increased number of c-Fos-positive cells in PaV, VMH and ARC nuclei in response to oleic acid compared to vehicle-treated mice. Conversely, ICV palmitic acid did not induce neuronal activation in these hypothalamic nuclei in comparison to the vehicle group, and this level of c-Fos expression was even lower when compared to oleic acid-treated mice (Figure 3).

### 2.4. Neuronal Activation Induced by Oleic Acid in the PaV and VMH Was Attenuated in the Presence of the Melanocortin 4 Receptor (MC4R) Antagonist

In order to explore the melanocortin pathway in the neuronal activation of the hypothalamic nuclei by central oleic acid, c-Fos expression was assessed in the absence or presence of the MC4R antagonist SHU9119 [10]. In the presence of SHU9119, oleic acid-induced neuronal activation was substantially attenuated in PaV and VMH sections of mice compared to the vehicle-treated group (Figure 4A,B). Neuronal activation in the ARC in response to ICV oleic acid remained unchanged in the presence of the melanocortin receptor antagonist (Figure 4A,B).

### 2.5. ICV Injection of Unsaturated versus Saturated FA Differently Affects AMPK/ACC Phosphorylation and Fatty Acid Synthase (FAS) Expression

Taking into account that the AMPK/ACC/malonyl-CoA/CPT1C has been involved in hypothalamic control of BAT thermogenesis in response to HFD, we first analyzed the effect of central LCFA administration on hypothalamic expression of the nutrient sensor AMPK and its downstream mediator ACC by immunoblotting. Among the LCFA, a different profile of the phosphorylated and total protein expression ratio was found. ICV oleic acid increased the pACC/ACC but decreased the pAMPK/AMPK ratio, whereas these ratios were not significantly changed in response to linoleic, palmitic and stearic acids (Figure 5A,B).

Another key player in the response of the hypothalamus to the nutritional state is FAS, which catalyzes fatty acid synthesis using malonyl-CoA as a substrate. The expression of FAS in the hypothalamus was reduced after linoleic acid injection and showed a clear tendency in response to oleic acid versus vehicle (Figure 5D,E). FAS expression was, in contrast, significantly upregulated in the hypothalamus of mice treated with saturated FA and palmitic and stearic acids (Figure 5D,E). These findings suggest that the unsaturated oleic and linoleic acids inhibit the FA synthesis pathway while saturated FA activate this pathway and may therefore affect malonyl-CoA fluctuations in the hypothalamus.

### 2.6. Mice Deficient in CPT1C Showed a Reduced Effect of ICV Oleic Acid on Body Weight, Thermogenesis and Hypothalamic Neuronal Activation

Considering the role of CPT1C in hypothalamic regulation of energy balance, the involvement of this neuronal protein in the satiating and thermogenic action of ICV oleic acid was investigated in CPT1C-KO. As illustrated in Figure 6, central injection of oleic acid was unable to reduce body weight and food intake change in CPT1C-KO mice (Figure 6A,B), in contrast to that observed in WT mice. In addition, CPT1C deficiency abrogated the action of central oleic acid in BAT thermogenic markers and iBAT temperature of CPT1C-deficient mice (Figure 6C–E).

Evaluation of the AMPK/ACC and FAS pathways revealed that ICV oleic acid also upregulated pACC/ACC expression ration in the hypothalamus of CPT1C-KO mice, but it was not able to downregulate pAMPK/AMPK ratio CPT1C-KO mice, as previously observed in WT mice (Figure 7A,B,D,E). Regarding the expression of FAS, hypothalamus of CPT1C-KO mice showed increased expression levels in response to ICV oleic acid (Figure 7C,F), in contrast to that observed in WT mice. In addition, analysis of neuronal activation indicated that central oleic acid was unable to increase c-Fos expression of the hypothalamic nuclei of CPT1C-KO mice in contrast to control mice. In particular, PaV, VMH and ARC were not activated or even showed a lower number of c-Fos positive cells by oleic acid in CPT1C-deficient mice compared to the vehicle-treated group (Figure 7G,H).

## 3. Discussion

Sensing and metabolism of lipids in the hypothalamus is a key signal of nutrient status to modulate energy balance, and its disruption leads to obesity and associated complications. In the hypothalamus, LCFA promote different types of intracellular signaling to regulate glucose homeostasis, appetite and energy expenditure. To date, the effects of saturated or unsaturated LCFA on the hypothalamus–BAT thermogenesis axis and the exact mechanisms underpinning these actions have remained largely unclear. The present research demonstrates that the presence of LCFA in the brain with different levels of saturation impacts differently on the regulation of appetite and BAT thermogenesis. We found notable BAT thermogenesis activation induced by a MUFA-enriched diet and central injection of unsaturated LCFA (oleic and linoleic acid), whereas this activation was not appreciated in response to saturated LCFA. The thermogenic action evoked by oleic acid implied the melanocortin system, FAS expression, and the neuron-specific protein CPT1C as a key mediator in lipid sensing.

Both quantity and quality of dietary fat, especially the presence and position of double bonds in FA, play critical roles in the regulation of diet-induced thermogenesis [17,35]. This evidence has been partly supported by a wide range of investigations exploring the metabolic actions of fish-oil derived ω3 PUFA, eicosapentanoic acid (EPA) and docosahexanoic acid (DHA), on adipose tissue phenotype [35,36,37]. However, although 18-carbon FAs represent the largest FA class in human diets, very few studies have been conducted to compare their relative effects on thermogenesis [38], and particularly on hypothalamic regulation of BAT function. We have demonstrated that administration of a HFD enriched in oleic acid for 7 days led to BAT thermogenesis activation with no changes in body weight gain, despite the increased calorie intake compared to control diet, whereas feeding a SFA-enriched diet resulted in higher body weight and lower activation of thermogenesis, in agreement with previous results [38]. The higher cumulative calorie intake observed with MUFA diet compared to SD or SFA diets could be explained by the palatability associated with this type of unsaturated diet [39]. Analysis of c-Fos immunofluorescence revealed an acute neuronal activation of specific areas of the hypothalamus (i.e., PaV) in response to a MUFA- but not a SFA-enriched diet, indicating the importance of hypothalamic signals on the thermogenic response induced by unsaturated FA-based diets.

To determine whether the distinct effects on BAT thermogenesis and neuronal activation observed with the MUFA-enriched diet were due to the presence of increased amounts of LCFA-CoA in the hypothalamus, as previously described for the anorexigenic effect [1,11], we next evaluated the direct effect of unsaturated (oleic acid 18:1 and linoleic acid 18:2) vs. saturated LCFA (stearic acid 18:0 and palmitic acid 16:0) in BAT thermogenesis and hypothalamic activation after ICV infusion. Central injection of oleic and linoleic acid activated BAT thermogenesis and reduced body weight and food intake, and this satiating effect was in agreement with previous investigations [1,10,40]. The importance of LCFA-dependent signaling in the hypothalamus to regulate feeding has been widely evaluated by different research groups, unlike BAT thermogenesis induction. Central administration of oleic acid reduced food intake and body weight over a 48 h period, whereas saturated LCFA did not affect or even raise feeding and body weight [1,10,40]. These findings suggest that the mere infusion of FA is not responsible, but rather that different signals involving saturation grade of LCFA are most likely involved. In line with these results, our study confirmed the satiating effect of ICV infusion of both oleic and linoleic acid, which was supported by an increased expression of the anorexigenic neuropeptide POMC in the hypothalamus, partly responsible for body weight attenuation, as previously demonstrated [10,40]. The obesogenic action of ICV saturated FA has been related to the promotion of inflammation [40,41], as observed in our study in terms of microglia activation by ICV palmitate infusion (Appendix A). The central inflammation led by saturated FA along with changes in the levels of specific intracellular metabolites (i.e., ceramides and phospholipids) in hypothalamic neurons could ultimately interfere in BAT activation and finally overweight [42].

In this study, we also reported that ICV infusion of oleic acid, but not palmitic acid, resulted in a higher number of c-Fos positive cells, indicating neuronal activation in the ARC, VMH and PaV sections of the hypothalamus. PaV has been suggested as a main responsible nucleus for oleic acid effects on energy balance through the melanocortin system POMC/MC4R [10,43]. Considering the remarkable neuronal activation of PaV and upregulation of POMC hypothalamic expression in response to ICV oleic acid, we investigated whether the melanocortin system could be involved in the oleic acid effects. The increase in c-Fos positive cells in both PaV and VMH induced by ICV oleic acid was blunted in the presence of SHU9119, an antagonist of the MC4R. Then, we could hypothesize that POMC neurons may sense the increased amount of oleic acid to in turn signal PaV and VMH nuclei through the melanocortin system, leading to changes in BAT thermogenesis and body weight.

Activation of BAT thermogenesis has been strongly demonstrated under the control of hypothalamic AMPK activity [44]. Attenuation of AMPK expression in the VMH resulted in BAT thermogenesis overactivation and body weight loss in a feeding-independent manner [45,46]. When AMPK is active, it phosphorylates and inhibits ACC followed by FAS expression attenuation and therefore modulation in malonyl-CoA levels. The dynamic on malonyl-CoA in the hypothalamus is a major nutrient-sensing pathway regulating food intake and energy expenditure [8,9]. In high energy conditions, hypothalamic malonyl-CoA levels rise, leading to food intake attenuation and enhanced energy expenditure, whereas attenuation in malonyl-CoA levels is a signal of feeding and energy expenditure reduction [8,9]. In response to ICV infusion of oleic acid, we observed a reduction in AMPK phosphorylation in the hypothalamus, which would be in concordance to increased BAT thermogenesis activation, whereas ACC phosphorylation levels increased, as previously reported [40]. The other LCFA tested in the study did not show significant differences in AMPK-ACC expression levels but instead led to changes in FAS expression. ICV injection of unsaturated LCFA reduced FAS expression in the hypothalamus, while saturated LCFA, on the contrary, upregulated it. These data indicate that FA synthesis is decreased by unsaturated FAs, and accumulation of the precursor malonyl-CoA is expected, leading to satiety and BAT thermogenesis activation, in line with previous studies [47]. The contrary would apply for saturated fats.

A major downstream effector of malonyl-CoA and AMPK in the hypothalamus to regulate diet-induced BAT thermogenesis is CPT1C [27]. Then, we explored the role of this neuron-specific protein as a molecular mediator in the thermogenic action of central oleic acid. The effects of ICV oleic acid in food intake, body weight and BAT thermogenesis activation were blunted in mice lacking CPT1C, suggesting that CPT1C is needed for central actions of oleic acid. We also studied the AMPK/ACC and FAS axis in these KO mice, resulting in pACC upregulation with no changes in pAMPK but increased FAS expression after oleic acid infusion, as observed in response to saturated FA in WT mice, indicating an attenuated malonyl-CoA signal in the hypothalamus. CPT1C has been proposed to be a sensor of malonyl-CoA levels [8,28,48], and its role in the energy homeostasis and metabolic adaptation depends on this FA intermediary [27,49]. Recent investigations have found that malonyl-CoA binding provides CPT1C the potential to interact and regulate the activity of other downstream proteins such as ABHD6 [50,51] and SAC1 [52], both involved in brain metabolic flexibility, and AMPA-type glutamate receptors trafficking to the cell surface in neurons [52], whereas this regulation is lost in conditions of malonyl-CoA depletion [50,52]. CPT1C was also essential for neuronal activation evoked by ICV oleic acid, since the number of c-Fos-positive cells was not increased but reduced in PaV and VMH in CPT1C-KO exposed to oleic acid. The interacting proteins of CPT1C can have a substantial impact on the excitability of neurons in the hypothalamus, which could explain the lack of neuronal activation associated with an impaired malonyl-CoA sensing by CPT1C in the hypothalamus in response to oleic acid to signal feeding and BAT thermogenesis regulation.

Altogether, central infusion of unsaturated but not saturated LCFA increased BAT thermogenesis and attenuated food intake, resulting in body weight loss. This effect on hypothalamic regulation of BAT thermogenesis involves changes in the AMPK/ACC/Malonyl-CoA axis in the hypothalamus, which is sensed by CPT1C to in turn drive melanocortin system activation in specific nuclei of the hypothalamus to ultimately promote BAT thermogenesis (Figure 8). These findings add new insight into the nutrient-activated neuronal circuitries crucial for the development and progression of metabolic diseases such as obesity and type 2 diabetes, and presents a new CPT1C-dependent mechanism underpinning central oleic acid-induced BAT thermogenesis.

## 4. Materials and Methods

### 4.1. Animals

Male (8–10 weeks old) CPT1C KO mice and their WT littermates with the same genetic background (C57BL/6J) were used for the experiments [53]. Mice were housed on 12 h/12 h light/dark cycle (light on at 8 a.m., light off 8 p.m.) in a temperature- and humidity-controlled room. Animals were allowed free access to water and standard laboratory chow diet, otherwise indicated. At the end of the experimental protocols, animals were sacrificed, and tissues were collected for further molecular and biochemical analysis as further detailed. All animal procedures were performed in agreement with European guidelines (2010/63/EU) and approved by the University of Barcelona Local Ethical Committee (CEEA-C-303/18 and CEEA-C-233/19 from the Generalitat de Catalunya).

### 4.2. ICV Cannulation Surgery

Chronic cannulae were stereotaxically implanted into the lateral cerebral ventricle under ketamine/xylazine intraperitoneal anesthesia (ketamine 75 mg/kg body weight plus xylazine 10 mg/kg body weight). The coordinates were 0.58 mm posterior to Bregma, 1 mm lateral to the midsagittal suture and to a depth of 2.2 mm. Mice were individually caged and allowed to recover for at least 5 days before the experiment. Prior to the experiment, cannula placement was verified by a positive dipsogenic response to angiotensin II (1 nmol in 1 mL; Sigma-Aldrich, Saint Louis, MO, USA). On experimental day, mice received an ICV administration of fatty acids or vehicle, as indicated in the experimental protocols section.

### 4.3. Experimental Protocols

#### 4.3.1. Administration of HFD

For these experiments, animals were randomly assigned to the following groups: mice fed a diet with high content in MUFA (MUFA diet, 49% kcal from fat, 34.6% kcal of MUFA, 7% kcal of SFA, ref. D19121203), a diet with high content in SFA (SFA diet, 49% kcal from fat, 34.8% kcal of SFA, 7.1% kcal of MUFA, ref. D19121204) or a standard diet (SD, 10% kcal from fat, 2.2% kcal of SFA, 2.8% kcal of MUFA, ref. D12450J, Research Diets, New Brunswick, NJ, USA). These experiments were performed:
HFD feeding for 7 days. During this period, body weight and the amount of each diet consumed was measured, and the caloric intake value was calculated. BAT temperature was also monitored. At the end of the experiment, mice were sacrificed by cervical dislocation, and hypothalamus and BAT were collected and stored at −80 °C until further processing.HFD feeding for 2 h. This protocol was designed to analyze neuronal activation in the hypothalamic nuclei. Mice were fasted for 3 h followed by a refeeding period of 2 h with MUFA, SFA or SD diet. After 2 h, mice were perfused for brain cryopreservation, as indicated below.

#### 4.3.2. ICV Administration of FA

ICV-cannulated mice were treated with unsaturated FA (oleic acid, linoleic acid) (O3880 and L8134, respectively; Sigma-Aldrich, St. Louis, MO, USA), saturated FA (palmitic acid, stearic acid) (P5177 and S3381, respectively; Sigma-Aldrich, St. Louis, MO, USA) or vehicle. Unsaturated FA were complexed with fresh 40% HPB (2-Hydroxypropyl)-β-cyclodextrin and saturated FA with 0.1 M NaOH, in 188 mg/mL of free FA bovine serum albumin (BSA). The final dose prepared was 4.5 mM. These protocols and dosages were adapted from others previously described [1,10,40]. The volumes injected were 2 µL/dose corresponding to 9 nmol/dose. The following approaches were carried out:
ICV injection of FA for 2 days (2 injections/day). Injections were performed 1 h after the light phase and 1 h before the dark phase. During this period, body weight, food intake and BAT temperature were monitored. At the end of the experiment, 2 h after the dark phase, were sacrificed by cervical dislocation and hypothalamus, and BAT were collected and stored at −80 °C until further processing.ICV injection of FA for 2 h. This protocol was designed to analyze neuronal activation in the hypothalamic nuclei. FA were injected and during this period; animals did not have access to food. After 2 h, mice were perfused for brain cryopreservation, as indicated below. In some experiments, mice were previously treated with the MC4R antagonist SHU9119 (M4603, Sigma-Aldrich, St. Louis, MO, USA) (ICV administration; 0.5 nmol/dose) with the FA injection. SHU9119 was prepared in aCSF as previously described [10].

### 4.4. BAT Temperature Measurements

iBAT skin temperature was measured using infrared technology (camera FLIR E95 24°, Teledyne FLIR Systems, Wilsonville, OR, USA) and analyzed with its specific software, FLIR TOOLS Thermography Software. Videos were recorded on the first and the last day of treatment. As previously described, skin surrounding iBAT was shaved under isoflurane (IsoFlo^®^, Zoetis, London, UK) anesthesia 2 days before recordings [27]. The increment of iBAT temperature was calculated as the difference of temperature from last day to first day of treatment.

### 4.5. RNA Preparation and Quantitative RT-PCR

Total RNA was extracted from tissues using Trizol Reagent (Fisher Scientific, Madrid, Spain). Retrotranscription and quantitative RT-PCR (qPCR) was performed as previously described [26]. The SYBR Green assay primers used are indicated in Appendix A (IDT DNA Technologies, Leuven, Belgium). Relative mRNA levels were measured using the CFX96 Real-time System, C1000 Thermal Cycler (BioRad, Hercules, CA, USA). Relative gene expression was estimated using the comparative Ct (2^−ΔΔCt^) method in relation to GAPDH levels.

### 4.6. Western Blotting

Western blot was performed as previously described [27]. Briefly, tissue was homogenized in RIPA buffer (Sigma-Aldrich, Madrid, Spain) containing protease and phosphatase inhibitor cocktails. Protein extracts were separated on SDS-PAGE, transferred into Immobilion-PVDF membranes (Merck Millipore, Madrid, Spain) and probed with antibodies against: ACC, AMPKα, pACC (Ser79), pAMPKα (Thr172), (Cell Signaling; Danvers, MA, USA); FAS (Santa Cruz, Dallas, TX, USA) and GAPDH (Abcam, Cambridge, UK) (Appendix A). Each membrane was then incubated with the corresponding horseradish peroxidase-conjugated secondary antibody, anti-mouse or anti-rabbit (Jackson, West Grove, PA, USA) (Appendix A), and developed using LuminataForte Western HRP substrate (Merck Life Sciences, Madrid, Spain ). Images were collected by the ChemiDoc MP and Image Lab Software (Bio-Rad Laboratories, Hercules, CA, USA) and quantified by densitometry using ImageJ-1.33 software (NIH, Bethesda, MD, USA). GAPDH was used as an endogenous control to normalize expression levels.

### 4.7. Brain Immunofluorescence

For brain fixation, mice were anesthetized under ketamine/xylazine and intracardially perfused with PBS and then 10% NBF at 2 h after the experimental protocol (diet exposure or ICV injection of fatty acids). Brains were collected and postfixated 24 h in 10% NBF at 4 °C, transferred to 30% sucrose at 4 °C for 2–3 days, frozen in isopentane, and sliced in 30 µm thick slices in the coronal plane throughout the entire rostral–caudal extent of the brain using a cryostat. The slices obtained were preserved at −20 °C in antifreeze solution until use.

For neuronal activation analysis (c-Fos), hypothalamic slices containing PaV, VMH, DMH or ACR nuclei were extensively washed in 0.1% Triton X-100 KPBS buffer and blocked in 2% goat anti-serum in KPBS and 3% BSA plus 0.1% Triton X-100. Sections were incubated with rabbit anti-c-Fos antibody (1:200, Cell Signaling, Danvers, MA, USA) in a blocking buffer for 1 h at room temperature. After washing with KPBS, slices were incubated with anti-rabbit Alexa Fluor 647 antibody (1:1000; Invitrogen, Waltham, MA, USA) for 1 h at room temperature, followed by counterstaining with Hoechst for nuclear staining (1 mg/mL, Sigma-Aldrich, Saint Louis, MI, USA). Representative images of co-immunostaining of c-Fos with Hoechst are shown in Appendix A.

For immunostaining of microglia, slices were incubated with rabbit anti-Iba1 antibody (1:500, WAKO, Richmond, VA, USA) (Appendix A), followed by the secondary antibody and Hoechst, as described. Preparations were mounted into Superfrost Plus Slides (J1800AMNZ, Thermo Fisher Scientific, Waltham, MA, USA) using antifade Fluoromount-G^®^ (0100-01, Southern Biotech, Birmingham, AL, USA) and coverslips. Images were taken using a Leica DMi8 confocal microscope and acquired with 10× (for c-Fos experiments), 20× and 40× objectives (for Iba-1 assays). Fluorescence integrated density after image masking was calculated using ImageJ FIJI (NIH) software (2.9.0).

### 4.8. Data Processing and Statistical Analysis

Prism 9.0 (Graphpad Software, San Diego, CA, USA) was used for data and statistical analyses. Data are expressed as mean ± SEM. Comparative analyses were performed using two tests depending on the number of groups aimed to compare. For only two groups, Student’s t test was used, while if the objective was to compare more than two groups between them and taking into account the whole experiment, the 2-way ANOVA test was used followed by post hoc two-tailed Bonferroni test. A p value less than 0.05 (*p* < 0.05) expressing differences between groups was considered statistically significant.

## Figures and Tables

**Figure 1 ijms-24-01697-f001:**
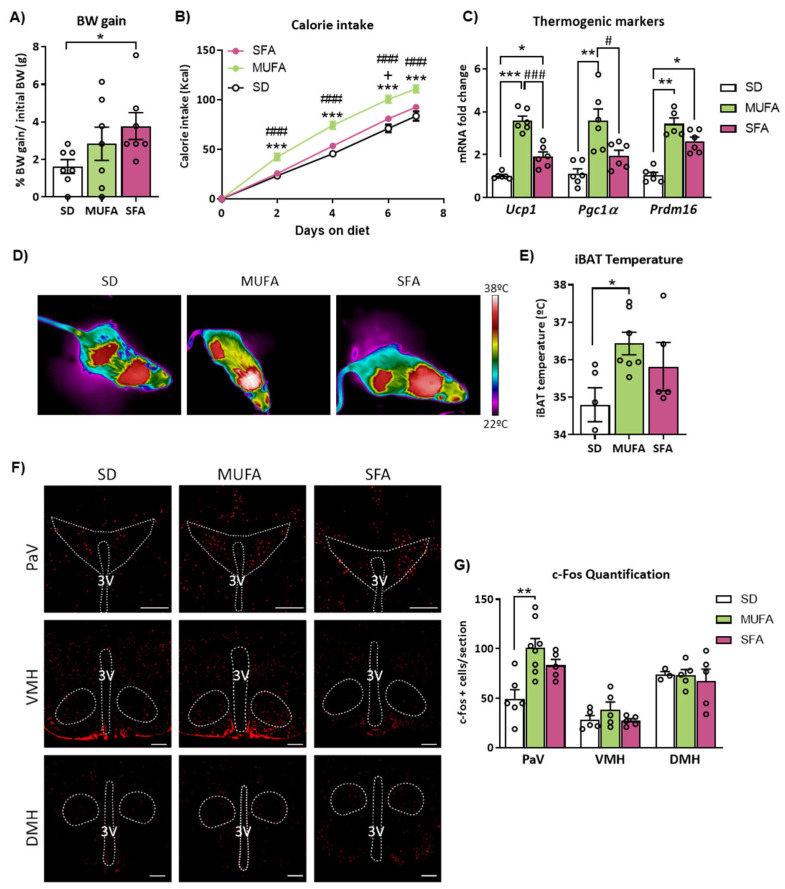
Effects of short-term administration of SD-, MUFA- and SFA-enriched diets in body weight, feeding, brown adipose tissue (BAT) thermogenesis and neuronal activation in the hypothalamus of WT mice**.** (**A**) Body weight (BW) gain normalized to the initial body weight measured, (**B**) cumulative caloric intake, (**C**) and mRNA expression of the thermogenic markers in BAT after 7 days on special diets. (**D**) Representative infrared pictures and (**E**) quantification of interscapular BAT (iBAT) temperature after 7 days on special diets. (**F**) Representative images and (**G**) quantification of immunohistochemistry for c-Fos expression in PaV, VMH and DMH after 2 h exposition to the special diets. Data are represented as mean ± SEM (n = 7 animals in body weight, food intake and thermogenesis activity measurement, n = 4 animals per group in c-Fos experiment). * *p* < 0.05, ** *p* < 0.01, *** *p* < 0.001 MUFA vs. SD; # *p* < 0.05, ### *p* < 0.001 MUFA vs. SFA; + *p* < 0.05 SFA vs. SD. MUFA: diet with high content in monounsaturated fatty acids. SFA: diet with high content in saturated fatty acids; SD: standard diet; PaV: paraventricular hypothalamus; VMH: ventromedial hypothalamus; DMH: dorsomedial hypothalamus; 3V: third ventricle. Scale bar = 250 µm.

**Figure 2 ijms-24-01697-f002:**
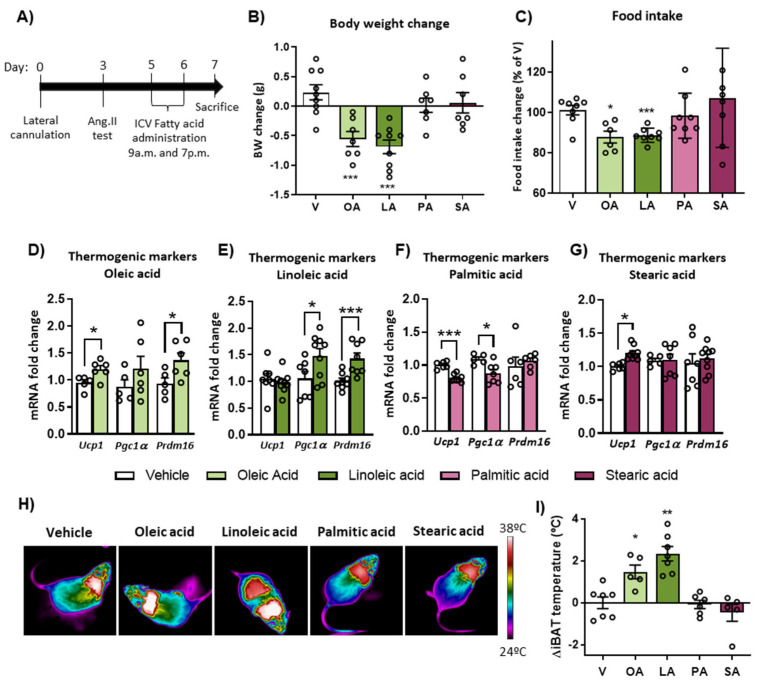
Effects of intracerebroventricular (ICV) administration of oleic acid, linoleic acid, palmitic acid and stearic acid on body weight, food intake and brown adipose tissue (BAT) thermogenesis in WT mice. (**A**) Schematic representation of the experimental protocol. (**B**) Body weight (BW) change and (**C**) cumulative food intake change measured during 48 h of treatment. (**D**–**G**) mRNA expression of the thermogenic markers UCP1, PGC1α and PRDM16 in BAT after 48 h ICV injection of the fatty acids. (**H**) Representative infrared thermal images and (**I**) quantification of interscapular BAT (iBAT) temperature change measured during the treatment. Data are represented as mean ± SEM (n = 5–9 per group). * *p* < 0.05, ** *p* < 0.01, *** *p* < 0.001 vs. vehicle. OA: oleic acid; LA: linoleic acid; PA: palmitic acid; SA: stearic acid; V: vehicle.

**Figure 3 ijms-24-01697-f003:**
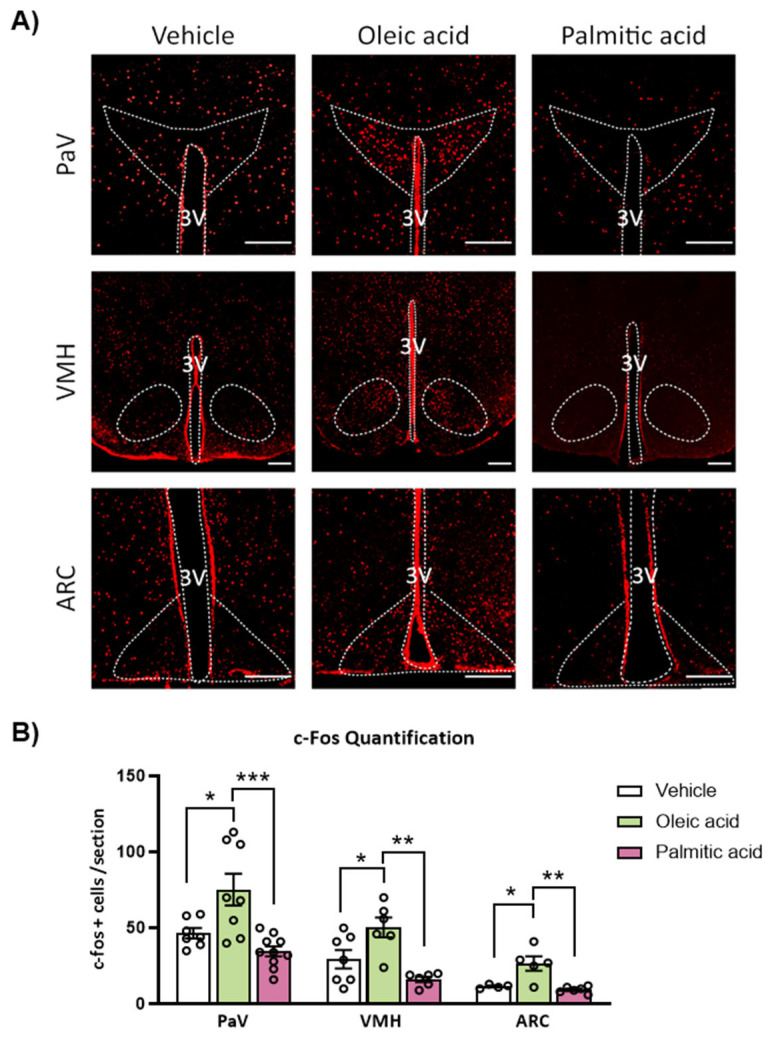
Effects of intracerebroventricular (ICV) administration of oleic acid and palmitic acid on neuronal activation on hypothalamic nuclei in WT mice. (**A**) Representative images and (**B**) quantification of immunohistochemistry for c-Fos expression in paraventricular (PaV), ventromedial (VMH) and arcuate (ARC) nucleus of the hypothalamus after 48 h of ICV administration of fatty acids. Data are represented as mean ± SEM (n = 3 per group; 2–3 slices/mice). * *p* < 0.05, ** *p* < 0.01, *** *p* < 0.001 vs. vehicle from same nucleus. OA: oleic acid; LA: linoleic acid; PA: palmitic acid; SA: stearic acid; V: vehicle. 3V: third ventricle. Scale bar = 250 µm.

**Figure 4 ijms-24-01697-f004:**
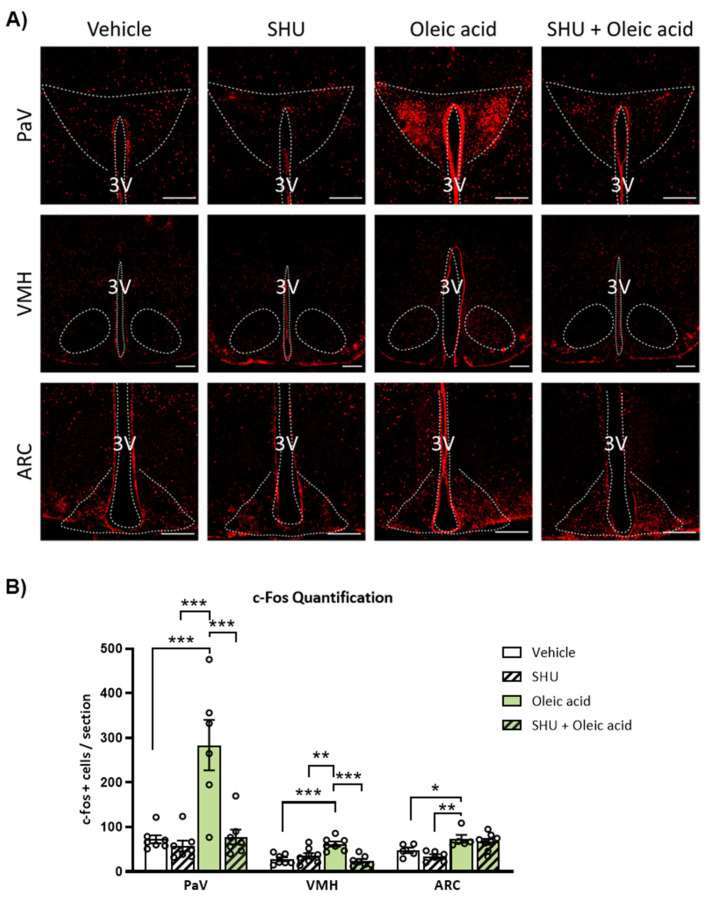
Effects of the intracerebroventricular (ICV) administration of the MC4R antagonist SHU9119 on neuronal activation induced by oleic acid in hypothalamic nuclei in WT mice**.** (**A**) Representative images and (**B**) quantification of immunohistochemistry for c-Fos expression in paraventricular (PaV), ventromedial (VMH) and arcuate (ARC) nucleus of the hypothalamus after 48 h of ICV administration of oleic acid. Data are represented as mean ± SEM (n = 3 per group; 2–3 slices/mice). * *p* < 0.05, ** *p* < 0.01, *** *p* < 0.001 vs. vehicle from same nucleus. 3V: third ventricle. Scale bar = 250 µm.

**Figure 5 ijms-24-01697-f005:**
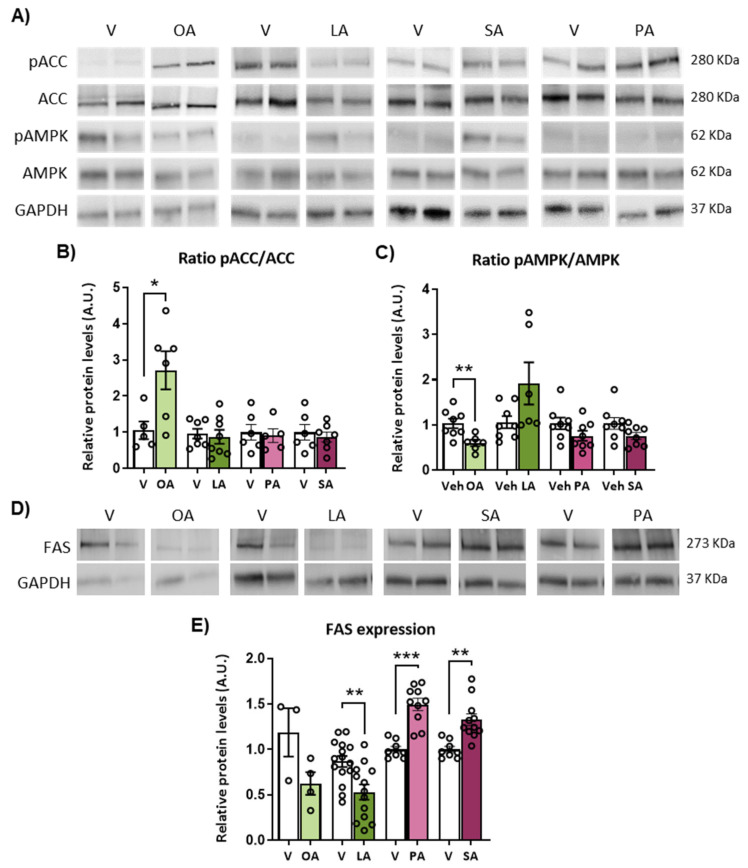
Effects of intracerebroventricular (ICV) administration of oleic acid, linoleic acid, palmitic acid and stearic acid on hypothalamic expression of ACC, AMPK and FAS in WT mice. (**A**) Representative blots of pACC, ACC, pAMPK and AMPK protein levels and (**B**,**C**) quantification of the pACC/ACC and pAMPK/AMPK ratios in the hypothalamus of mice after 48 h of ICV administration of fatty acids. (**D**) Representative blots and (**E**) quantification of FAS protein levels in the hypothalamus of mice after 48 h of ICV administration of fatty acids. Data are represented as mean ± SEM (3 blots from 5–7 different mice per group). * *p* < 0.05, ** *p* < 0.01, *** *p* < 0.001 vs. vehicle. FAS: Fatty acid synthase. OA: oleic acid; LA: linoleic acid; PA: palmitic acid; SA: stearic acid; V: vehicle.

**Figure 6 ijms-24-01697-f006:**
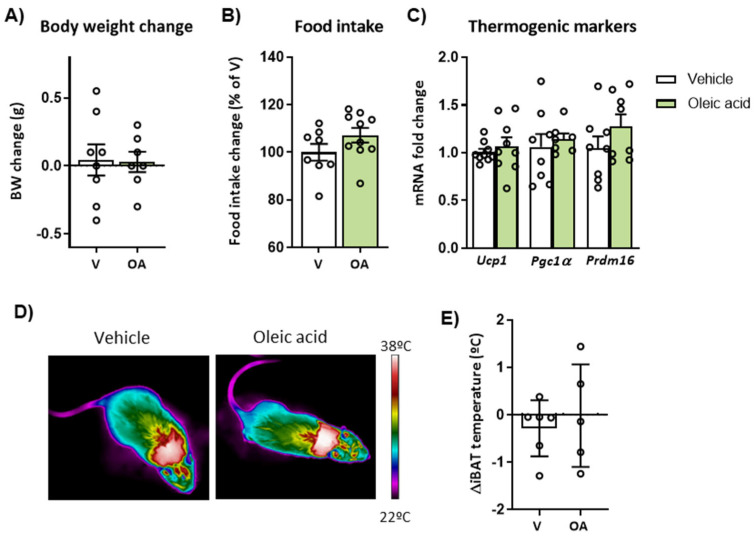
Effects of the intracerebroventricular (ICV) administration of oleic acid on body weight, food intake and brown adipose tissue (BAT) thermogenesis in CPT1C-KO mice. (**A**) Body weight (BW) change and (**B**) cumulative food intake measured during 48 h of treatment. (**C**) mRNA expression of the thermogenic markers UCP1, PGC1α and PRDM16 in BAT after 48 h ICV injection of oleic acid. (**D**) Representative infrared thermal images and (**E**) quantification of interscapular BAT (iBAT) temperature changes measured during the treatment. Data are represented as mean ± SEM (n = 7–9 per group for **A**–**C** panels and n = 5–6 per group for **D**,**E** panels). OA: oleic acid; V: vehicle.

**Figure 7 ijms-24-01697-f007:**
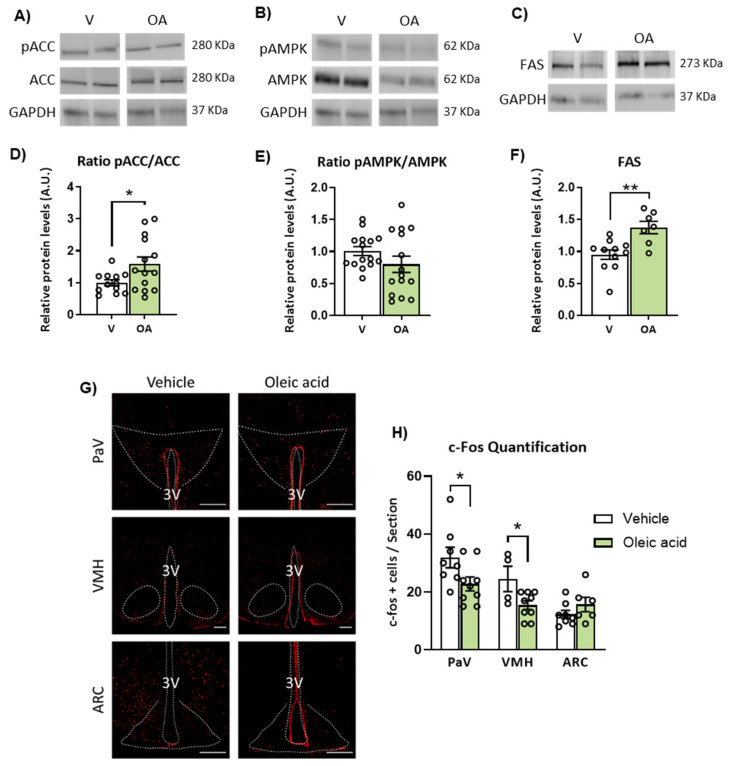
Effects of intracerebroventricular (ICV) administration of oleic acid on the expression of ACC, AMPK and FAS and neuronal activation in the hypothalamus of CPT1C-KO mice. (**A**–**C**) Representative blots of pACC, ACC, pAMPK, AMPK and FAS and (**D**–**F**) quantification of the pACC/ACC and pAMPK/AMPK ratios and FAS protein levels in the hypothalamus of mice after 48 h of ICV administration of oleic acid. (**G**) Representative images and (**H**) quantification of immunohistochemistry for c-Fos expression (c-Fos positive cells) in paraventricular (PaV), ventromedial (VMH) and arcuate (ARC) nucleus of the hypothalamus after 48 h of ICV administration of oleic acid. Data are represented as mean ± SEM (panels **D**–**F**: 2 blots from 8 different mice per group; panel H: n = 3 per group, 2–3 slices/mice). * *p* < 0.05, ** *p* < 0.01 vs. its corresponding vehicle. FAS: Fatty acid synthase. OA: oleic acid; V: vehicle. 3V: third ventricle. Scale bar = 250 µm.

**Figure 8 ijms-24-01697-f008:**
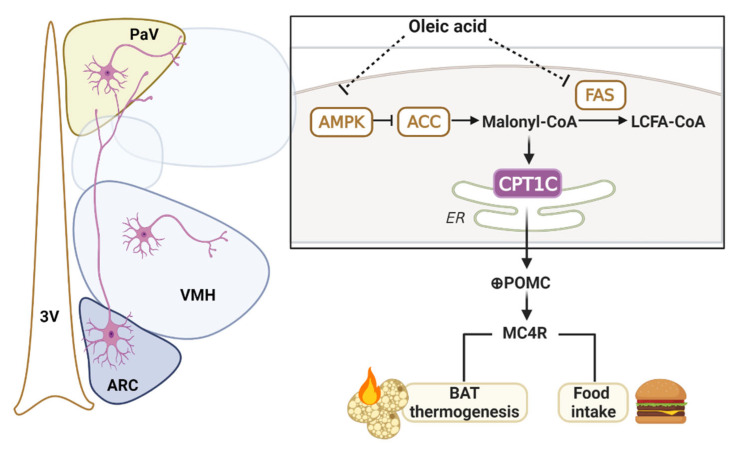
Central infusion of the unsaturated LCFA oleic acid, but not saturated LCFA, induced BAT thermogenesis and attenuate food intake, resulting in body weight loss. This action of oleic acid on BAT thermogenesis involves changes in AMPK/ACC/Malonyl-CoA and FAS in the hypothalamus, which is sensed by CPT1C to in turn drive melanocortin system activation in specific nuclei of the hypothalamus. ARC: arcuate nucleus of the hypothalamus; BAT: brown adipose tissue; FAS: fatty acid synthase; LCFA: long-chain fatty acid; PaV: paraventricular hypothalamus; VMH: ventromedial hypothalamus.

## Data Availability

Not applicable.

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
