# Peer review of "Central Regulation of Brown Fat Thermogenesis in Response to Saturated or Unsaturated Long-Chain Fatty Acids"

_ijms, 2023, doi:10.3390/ijms24021697_

Round 1

Reviewer 1 Report

This manuscript concerns regulation of brown fat thermogenesis by dietary changes, particularly increase of amount of saturated or unsaturated long-chain fatty acids (LCFA) in animal (mice) model. Based on previous data the authors decided to evaluate hypothalamic-brown adipose tissue (BAT) axis regulation. Central infusion of unsaturated but not saturated LCFA increased BAT thermogenesis and attenuate food intake, resulting in body weight loss, where regulation of BAT thermogenesis involves changes in AMPK/ACC/Malonyl-CoA axis in the hypothalamus. Carnitine palmitoyltransferase 1C is crucial for this control.

Manuscript by Fosch et al. is interesting and very important in the field of metabolic diseases, like diabetes (type 2) or obesity. In general, I am suggesting only some editorial corrections:

1. Please use full word in case of Figures, e.g. Fig. 1 --> Figure 1; Suppl. Fig. S2 --> Supplementary Figure S2

2. Figure 1B is missing legend for colors of curves.

3. Please, use italics for genes on mRNA level of expression, e.g. Figure 1C, 2D-G, 6C, S2, section 4.5. GAPDH.

4. Descriptions of Supplementary Figures should be under plots.

5. All abbreviations should be described when first time mentioned in the main text and every time in Figures description, e.g. PaV, VMH, DMH, ICV.

6. Please correct- Section 2.3. acces --> access AND section 4.3.2. ul --> uL

Reviewer 2 Report

The manuscript entitled “Central regulation of brown fat thermogenesis in response to saturated or unsaturated long-chain fatty acids” by Anna Fosch et al., was presented as an original Article in which the authors analyze the effects of unsaturated LCFA oleic acid on BAT thermogenesis and the attenuation of food intake. The research work is comprehensive and systematic and the text is well written, in a clear and organized manner. There are several major issues that need to be addressed/explained before considering publication in IJMS: 

Major comments for the text: 

It would be appropriate modifying the “Animals” section (Paragraph 4.1.) in Materials and Methods. The sentence “C57BL/6J (wild-type, WT) and CPT1C-KO male mice(8-10 weeks old) were used for the experiments” (page 13, lines 424-425) is confusing. It may seem to the reader that the authors have compared C57BL/6J mice versusCPT1C-KO mice. It is more appropriate the sentence “Male (8-10 weeks old) CPT1C KO mice and their wild-type (WT) littermates with the same genetic background (C57BL/6J) were used for the experiments” reported in Rodríguez-Rodríguez, R. et al., Molecular Metabolism. 2018 doi:10.1016/j.molmet.2018.10.010Furthermore, to specify the origin of the CPT1C-KO micethe reviewer suggests replacing the reference [27] with Carrasco Pet al., J Biol Chem. 2012 doi: 10.1074/jbc.M111.337493, in which is correctly reported the construction of targeting pPNT-CPT1C-KO vector and the generation of knock-out mice.

Please, double check and correct the typos of words/acronyms/abbreviations/tenses in the text. The reviewer suggests: 

a) to capitalize the first letters of full words that define the acronyms/abbreviations:

long-chain fatty acids (LCFA)brown adipose tissue (BAT)endoplasmic reticulum (ER)FA oxidation (FAO)high-fat diet (HFD)monounsaturated FA (MUFA)saturated FA (SFA)standard diet (SD)Body weight (BW)fatty acid synthase (FAS)OA: oleic acidLA: linoleic acidPA: palmitic acidSA: stearic acidV: vehicledocosahexanoic acid (DHA)

b) to specify and introduce an acronym in parentheses after the written-out form (i.e. High-Fat Diet, HFD in the abstract section) c) once you have entered an acronym, make sure that you no longer repeat the extended form in the text. d) It must be possible to read the abstract as a stand-alone work. Therefore, all abbreviations must already be defined in the abstract, such as ICV infusionPaV, VMN and ARC. e) It is necessary to implement some sections adding specific references: (i.e. “MC4R antagonist SHU9119, paragraph 2.4., page 6, line 208; SHU9119, an antagonist of the MC4R”, Discussion, page 12, line 357).

Major comments for the data:

Please, provide the original uncropped and unprocessed scans of all blots in the section of Original Images for Blots/GelsThis should be cited once in the Materials and Methods section (4.6. Western blotting)

The reviewer suggests creating a table of antibodies used for immunofluorescence and Western Blot analyses, including the catalog number code, the concentration, and the hybridization conditions and a table of primers used for quantitative RT-PCR.

Major comments for the figures:

Although the staining for c-Fos expression are convincing, it would be appropriate showing DAPI staining in parallel to show the correct cellular localization of signal. In addition, specify the magnification (x) in the Figure Legends
